# Disease progression in osteosarcoma: a multistate model for the EURAMOS-1 (European and American Osteosarcoma Study) randomised clinical trial

Audinga-Dea Hazewinkel [ID],[1,2,3,4] Carlo Lancia,[5] Jakob Anninga,[6] Michiel van de Sande,[7] Jeremy Whelan [ID],[8] Hans Gelderblom,[4] Marta Fiocco[3,5]

For numbered affiliations see end of article.

**Correspondence to**
Audinga-Dea Hazewinkel;
a.d.hazewinkel@bristol.ac.uk

## ABSTRACT

**Objectives** Investigating the effect of prognostic factors in a multistate framework on survival in a large population of patients with osteosarcoma. Of interest is how prognostic factors affect different disease stages after surgery, with stages of local recurrence (LR), new metastatic disease (NM), LR+NM, secondary malignancy, a second NM, and death.

**Design** An open-label, international, phase 3 randomised controlled trial.

**Setting** 325 sites in 17 countries.

**Participants** The subset of 1631 metastases-free patients from 1965 patients with high-grade resectable osteosarcoma, from the European and American Osteosarcoma Study.

**Main outcome measures** The effect of prognostic factors on different disease stages, expressed as HRs; predictions of disease progression on an individual patient basis, according to patient-specific characteristics and history of intermediate events.

**Results** Of 1631 patients, 526 experienced an intermediate event, and 305 died by the end of follow-up. An axial tumour site substantially increased the risk of LR after surgery (HR=10.84, 95% CI 8.46 to 13.86) and death after LR (HR=11.54, 95% CI 6.11 to 21.8). A poor histological increased the risk of NM (HR=5.81, 95% CI 5.31 to 6.36), which sharply declined after 3 years since surgery. Young patients (<12 years) had a lower intermediate event risk (eg, for LR: HR=0.66, 95% CI 0.51 to 0.86), when compared with adolescents (12–18 years), but had an increased risk of subsequent death, while patients aged >18 had a decreased risk of death after event (eg, for death after LR: HR=2.40, 95% CI 1.52 to 3.90; HR=0.35, 95% CI 0.21 to 0.56, respectively).

**Conclusions** Our findings suggest that patients with axial tumours should be monitored for LR and patients with poor histological response for NM, and that for young patients (<12) with an LR additional treatment options should be investigated.

**Trial registration number** NCT00134030.

## INTRODUCTION

Osteosarcoma is the most common primary bone sarcoma, with a primary peak incidence in adolescents and young adults and a second peak in patients of 50 years and older, often due to underlying conditions.[1] Current management strategies include neoadjuvant chemotherapy and surgical removal of the primary tumour and, if resectable, all metastatic disease.[2]

The European and American Osteosarcoma Study (EURAMOS-1) study (NCT00134030) was headed by the EURAMOS collaboration and recruited a total of 2260 patients from 2005 to 2011.[3] The relationship between various predictors and event-free and overall survival (OS) has been investigated previously.[4–6] Such analyses, however, only consider one (composite) outcome at the same time, and cannot take into account disease evolution, and how patient history affects the final prognosis.

Here, we reanalyse the EURAMOS-1 data using a multistate model.[7] A conventional Cox proportional hazards regression defines a single starting state (eg, study entry) and final state (eg, death). A multistate model extends this by introducing intermediate states a patient may transition to (eg, the development of local recurrence or metastatic disease). The aim of this study is to investigate

the effect of prognostic factors on different disease stages, and predict disease progression on an individual patient basis, estimating the probability of occupying a given disease stage according to patient-specific characteristics and history of intermediate events.

### Strengths and limitations of this study

► A multistate model takes into account patient history and estimates the effect of prognostic factors on the hazard of transitioning to and from specific events over the course of disease progression.

► A multistate model can be used to quantify the risk of experiencing specific disease stages, conditional on prior events and patient characteristics.

► A multistate model can be used to calculate individual, patient-specific probabilities of occupying a given disease stage.

► Multistate model estimation is limited to transitions with sufficient events per predictor category.

## PATIENTS AND METHODS
### Patients

The EURAMOS-1 trial data, per November 2014, contain information on 2260 patients, and include predictor measurements and records of intermediate events observed during follow-up.[3] Resection of the primary tumour was performed post neoadjuvant chemotherapy. Following surgery, 1136 of 2260 patients were randomised to treatment, subject to the histological response as assessed in the resected specimen. Patients with a poor response (≥10% viable tumour) were allocated MAP (methotrexate, doxorubicin, cisplatin) or MAP with ifosfamide and etoposide, while patients with a good response (<10%) received MAP or MAP followed by pegylated interferon. The primary analysis found no beneficial effect of experimental treatment in either group.[4][5] Therefore, we included both randomised and non-randomised patients in our analysis. To ensure valid inference, we selected a homogeneous subset of the data, excluding patients with a non-resectable primary tumour and patients with clinically detectable metastatic disease prior to surgery, as the latter comprise a biologically distinct population with a much poorer prognosis. A total of 1631 patients were considered to be eligible for analysis (figure 1). Follow-up was defined from surgery, with a maximum follow-up time of 9 years and a median of 5 years. Six variables were selected, which have previously been examined in the context of overall and event-free survival (EFS).[6] Table 1 shows the distribution of patients across predictor categories. No major differences were observed between randomised and non-randomised patients.

### Statistical analysis
#### Multistate model

In the EURAMOS data, information on various postsurgical events was recorded, with up to two intermediate events reported per patient. Online supplemental figure 1 (Appendix A) gives an overview of all events and transitions recorded in the data. We selected all intermediate states and transitions that had sufficient events to ensure reliable estimation. We distinguish the following intermediate events: local recurrence (LR), new metastatic disease (NM), the combination of both (LR +NM) and secondary malignancy (SM). NM includes new pulmonary metastases (60%), bone metastases (7%), metastases at other sites (19%) and any combination of the three (14%).

In our model, surgical resection of the primary tumour was chosen as a starting state, with 1631 patients. Of these, 526 experienced an intermediate event, moving to the states of either LR (n=61), NM (n=407), LR +NM (n=35), or SM (n=23), with 280 patients transitioning to death afterwards. An additional 25 patients died without experiencing an

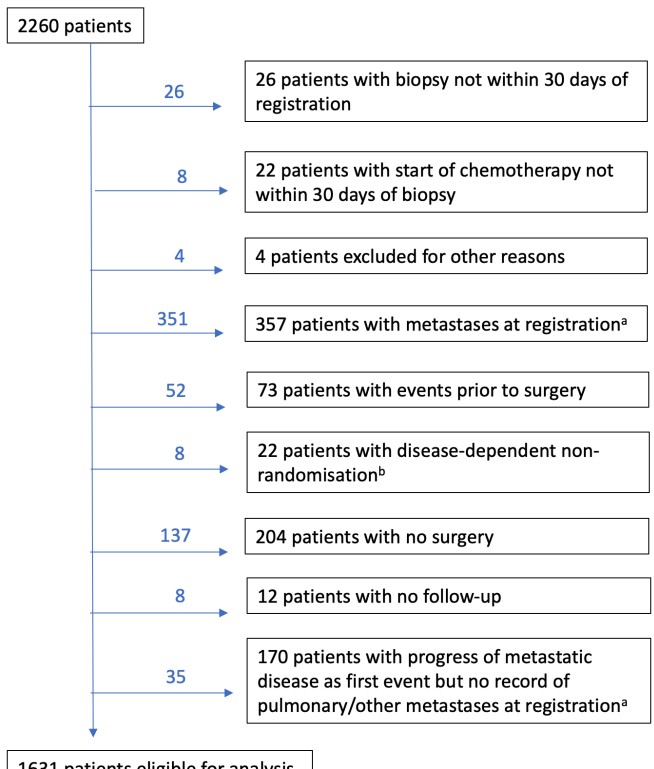

**Figure 1** Consolidated Standards of Reporting Trials diagram of patients included in the analysis. In the boxes, we list the six exclusion criteria with the total number of patients per category. Above the arrows, we give the additional number of patients excluded on considering each criterion, in order of appearance. (a) To ensure a homogeneous study population, we excluded patients with metastases prior to surgery. For 357 patients, metastases were recorded at registration, while for 170 patients, progression of new metastatic disease was found after surgery, while no metastases were recorded at registration. These patients were retrospectively reclassified as having metastatic disease prior to surgery and excluded from the analysis; (b) 22 randomised patients were later found to be ineligible due to progression of metastatic disease or new metastatic disease (n=11), or primary and/or metastatic unresectable disease (n=11).

intermediate event. Of the 407 patients experiencing an NM, 55 experienced a second metastatic disease (NM2) after remission, with 33 patients dying subsequently.

Figure 2 shows a schematic representation of the fitted multistate model, which consists of seven states and 10 transitions. Appendix A, with online supplemental figures 1–3, provides additional detail on model choice and the definition of states and transitions.

### Time varying effect

The model was stratified on transition, allowing a separate baseline for each of the seven transitions. Histological response violated the proportional hazards assumption for the transitions from surgery to NM and from NM to death, indicating that the hazard did not remain constant over time.

**Table 1** Patient demographics and disease characteristics

| Predictor | Treatment randomisation | | | | | |
| | Randomised | | Not randomised | | Total | |
| | N | % | N | % | N | % |
|---|---|---|---|---|---|---|
| **Age** | | | | | | |
| 12–18 | 627 | 38 | 275 | 17 | 902 | 55 |
| <12 | 276 | 17 | 117 | 7 | 393 | 24 |
| >18 | 219 | 13 | 117 | 7 | 336 | 21 |
| Missing | 0 | 0 | 0 | 0 | 0 | 0 |
| **Histological response*** | | | | | | |
| Good (<10% tumour) * | 614 | 38 | 231 | 14 | 845 | 52 |
| Poor (≥10% tumour) | 505 | 31 | 240 | 15 | 745 | 46 |
| Missing | 3 | 0 | 38 | 2 | 41 | 3 |
| **Excision** | | | | | | |
| Wide/radical | 923 | 57 | 404 | 25 | 1327 | 81 |
| Marginal | 149 | 9 | 55 | 3 | 204 | 13 |
| Intralesional Other | 15 | 1 | 7 | 0 | 22 | 1 |
| Unknown† | 11 | 1 | 5 | 0 | 16 | 1 |
| Missing | 24 | 1 | 38 | 2 | 62 | 4 |
| **Volume‡** | | | | | | |
| <200 | 633 | 39 | 303 | 19 | 936 | 57 |
| ≥200 | 265 | 16 | 126 | 8 | 391 | 24 |
| Missing | 224 | 14 | 80 | 5 | 304 | 19 |
| **Sex** | | | | | | |
| Female | 470 | 29 | 220 | 13 | 690 | 42 |
| Male | 652 | 40 | 289 | 18 | 941 | 58 |
| Missing | 0 | 0 | 0 | 0 | 0 | 0 |
| **Tumour location§** | | | | | | |
| Other | 956 | 59 | 413 | 25 | 1369 | 84 |
| Proximal femur/humerus | 133 | 8 | 74 | 5 | 207 | 13 |
| Axial | 33 | 2 | 22 | 1 | 55 | 3 |
| Missing | 0 | 0 | 0 | 0 | 0 | 0 |

*A good and poor histological response are defined by the amount of tumour remaining after resection, with <10% and ≥10% constituting a good and poor response, respectively.
†A subset of patients had an excision marked as 'unknown'. In the analysis, this category is treated as missing data and imputed.
‡Absolute volume is measured in cm × cm × cm × 0.54 (spheric T vol), or 0.785 for cylindric T vol).
§Tumour location was defined in accordance with the definition used in the Smeland *et al*. (2019) analysis of survival and prognosis in the EURAMOS-1 trial. Information was pooled from study variables 'site' (eg, femur, pelvis, spine, etc) and 'location' (eg, proximal, axial, etc). Observed axial tumour locations included rib (14) and pelvis/sacrum (41).
EURAMOS-1, European and American Osteosarcoma Study.

To account for the time-varying effect, an interaction term of histological response with the exponent of time was included.

### Missing data

Of the 1631 patients, 1264 patients were complete cases, with the greatest missingness observed for absolute tumour volume (19%). To make full use of the available data, missing values were imputed in a 10-fold imputation approach. Pooled estimates of the coefficients and SD were obtained using Rubin's rule.[8] For the predictor surgical excision, an 'unknown' excision was reported for 16 patients. A sensitivity analysis was conducted by evaluating estimate consistency across three different models,

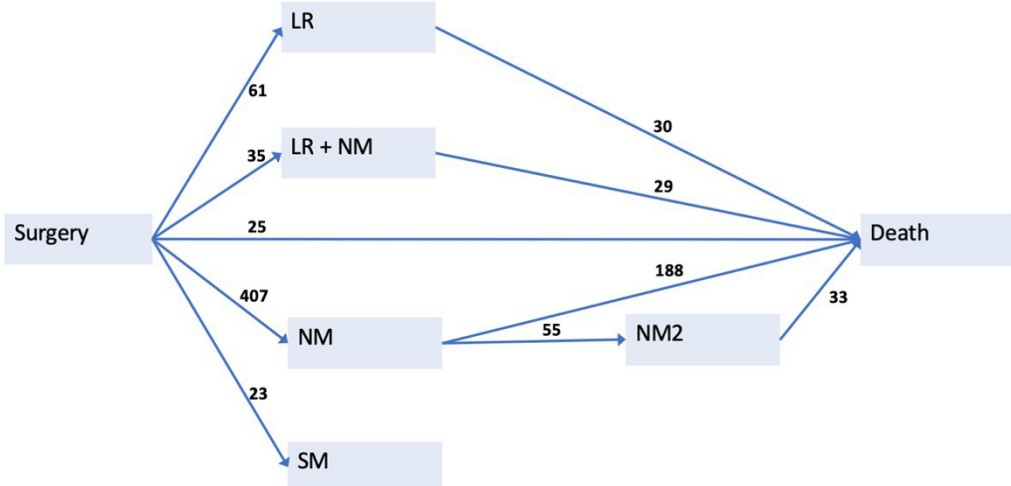

**Figure 2** Disease progression of osteosarcoma represented in a multistate model. Seven possible states and 10 transitions are defined. For each transition, the number of patients progressing from one state to another is shown. A total of 1631 patients are present in the starting stage, Surgery. After surgery, a patient may experience a local recurrence (LR), an LR+new metastatic disease (LR +NM), an NM, a secondary malignancy (SM) or death. After any such intermediate event, a patient may progress to death. Patients with NM may experience a second new metastatic disease after remission (NM2).

with unknown excision patients excluded from the data, with unknown excision modelled as a separate predictor category, and with unknown excisions treated as missing data and imputed. In the absence of substantial differences in model estimates, the latter option was selected.

The statistical analysis was performed in the R-software environment (R V.3.4.2),[9] with the *mstate*[10] and *Amelia*[11] libraries.

### Patient and public involvement

Patients or the public were not involved in the design, conduct, reporting or dissemination plans of our research (a reanalysis of the EURAMOS-1 trial using a multistate modelling approach).

### RESULTS
### Transition-specific HRs

The multistate model is illustrated in figure 2, along with the number of events. HRs with 95% CIs were estimated with a multivariate Cox proportional hazard regression model, modelling all transitions with sufficient events (online supplemental table 1, Appendix A). Tables 2 and 3 show the estimates for all transitions from surgery to intermediate event, and for all transitions terminating in death, respectively.

An axial tumour site, compared to any other limb site except proximal, was associated with a highly increased risk of LR after surgery (HR=10.84, 95% CI 8.46 to 13.86, table 2), death after LR (HR=11.54, 95% CI 6.11 to 21.78, table 3), and event-free death, that is, patients who move directly from surgery to death (HR=6.92, 95% CI 4.50 to 10.65, table 3), and with a moderate increased risk of death after NM (HR=1.34, 95% CI 1.02 to 1.76, table 3). Patients with an axial tumour were comparatively less

likely to experience an NM after surgery (HR=0.70, 95% CI 0.56 to 0.87, table 2).

Proximal tumour location was associated with moderately increased risks of LR (HR=1.60, 95% CI 1.21 to 2.12), NM (HR=1.41, 95% CI 1.28 to 1.56), event-free death (HR=2.30, 95% CI 1.59 to 3.34), death after LR (HR=2.72, 95% CI 1.70 to 4.35), and death after NM (HR=1.33, 95% CI 1.15 to 1.53) .

A poor histological response was associated with an increased risk of NM (HR=5.81, 95% CI 5.31 to 6.36), LR +NM (HR=3.82, 95% CI 2.86 to 5.10), SM (HR=2.07, 95% CI 1.48 to 2.90) and LR (HR=1.76, 95% CI 1.44 to 2.13). Poor histological response was also associated with an increased risk of event-free death (HR=1.55, 95% CI 1.15 to 2.10), and an increased risk of death in patients who had experienced LR +NM (HR=2.45, 95% CI 1.56 to 3.86), only LR (HR=1.75, 95% CI 1.24 to 2.47), or NM (HR=1.47, 95% CI 1.29 to 1.70).

An intralesional surgical excision, compared with a wide or radical one, was strongly associated with an increased risk of a second NM (HR=4.10, 95% CI 2.61 to 6.43), death after experiencing a second NM (HR=5.30, 95% CI 2.86 to 9.29), and more modestly with LR after surgery (HR=2.08, 95% CI 1.32 to 3.29). Patients with either an intralesional or marginal surgical excision were less likely to progress from LR to death (HR=0.33, 95% CI 0.16 to 0.67; HR=0.38, 95% CI 0.22 to 0.75, respectively).

Younger patients aged <12, compared with adolescents aged 12–18, had a decreased risk of LR (HR=0.66, 95% CI 0.51 to 0.86), NM (HR=0.80, 95% CI 0.72 to 0.88), SM (HR=0.61, 95% CI 0.42 to 0.89), and event-free death (HR=0.56, 95% CI 0.37 to 0.84). However, younger patients had an increased risk of death after LR (HR=2.40, 95% CI 1.52 to 3.90), LR+NM (HR=2.84, 95% CI 1.83 to 4.41), and NM2 (HR=2.34, 95% CI 1.69 to 3.22). Patients

**Table 2** HR estimates from multistate model per transition, for all transitions from surgery to intermediate event

| Predictor | Surgery → local recurrence | | Surgery → new metastatic disease | | Surgery → new metastatic disease +local recurrence | | Surgery → secondary malignancy | | New metastatic disease → new metastatic disease 2 | |
|---|---|---|---|---|---|---|---|---|---|---|
| | HR | 95% CI | HR | 95% CI | HR | 95% CI | HR | 95% CI | HR | 95% CI |
| **Age** | | | | | | | | | | |
| 12–18 | 1 | | 1 | | 1 | | 1 | | 1 | |
| <12 | 0.662 | 0.509 to 0.861 | 0.798 | 0.724 to 0.88 | 0.844 | 0.612 to 1.165 | 0.612 | 0.42 to 0.893 | 1.024 | 0.793 to 1.323 |
| >18 | 0.868 | 0.688 to 1.094 | 1.088 | 0.996 to 1.187 | 0.933 | 0.689 to 1.265 | 0.575 | 0.363 to 0.91 | 0.672 | 0.519 to 0.869 |
| **Histological response** | | | | | | | | | | |
| Good (<10% tumour) | 1 | | 1 | | 1 | | 1 | | 1 | |
| Poor (≥10% tumour) | 1.755 | 1.444 to 2.133 | 5.814 | 5.314 to 6.361 | 3.818 | 2.856 to 5.104 | 2.072 | 1.479 to 2.904 | 1.052 | 0.859 to 1.289 |
| Time effect | | | 0.986 | 0.985 to 0.987 | | | | | | |
| **Excision** | | | | | | | | | | |
| Wide/radical | 1 | | 1 | | | | | | 1 | |
| Marginal | 0.779 | 0.587 to 1.033 | 1.096 | 0.985 to 1.219 | | | | | 0.862 | 0.625 to 1.189 |
| Intralesional | 2.082 | 1.32 to 3.285 | 1.212 | 0.938 to 1.566 | | | | | 4.095 | 2.609 to 6.428 |
| **Location** | | | | | | | | | | |
| Other | 1 | | 1 | | | | | | | |
| Proximal femur/ humerus | 1.601 | 1.208 to 2.123 | 1.410 | 1.278 to 1.556 | | | | | | |
| Axial | 10.837 | 8.458 to 13.885 | 0.700 | 0.564 to 0.869 | | | | | | |
| **Sex** | | | | | | | | | | |
| Female | 1 | | 1 | | 1 | | 1 | | 1 | |
| Male | 0.947 | 0.783 to 1.145 | 1.048 | 0.972 to 1.13 | 1.402 | 1.079 to 1.82 | 0.724 | 0.534 to 0.982 | 1.343 | 1.092 to 1.652 |
| **Volume** | | | | | | | | | | |
| <200 | 1 | | 1 | | 1 | | 1 | | 1 | |
| ≥200 | 0.802 | 0.642 to 1.001 | 1.513 | 1.397 to 1.638 | 1.499 | 1.152 to 1.951 | 0.755 | 0.502 to 1.134 | 0.760 | 0.592 to 0.975 |

Bold values indicate hazard ratios (HRs) with 95% confidence intervals (CIs) that exclude the null.

**Table 3** HR estimates from multistate model per transition, for all absorbent transitions from intermediate event to death

| Predictor | Surgery → death | | Local recurrence → death | | New metastatic disease → death | | New metastatic disease +local recurrence→death | | New metastatic disease 2 → death | |
|---|---|---|---|---|---|---|---|---|---|---|
| | HR | 95% CI | HR | 95% CI | HR | 95% CI | HR | 95% CI | HR | 95% CI |
| Age | | | | | | | | | | |
| 12–18 | 1 | | 1 | | 1 | | 1 | | 1 | |
| <12 | **0.555** | **0.368 to 0.836** | **2.401** | **1.517 to 3.799** | 0.881 | 0.757 to 1.024 | **2.838** | **1.825 to 4.413** | **2.335** | **1.694 to 3.218** |
| >18 | 0.739 | 0.506 to 1.081 | **0.345** | **0.213 to 0.560** | 0.892 | 0.785 to 1.014 | **0.473** | **0.314 to 0.711** | **0.529** | **0.338 to 0.829** |
| Histological response | | | | | | | | | | |
| Good (<10% tumour) | 1 | | 1 | | 1 | | 1 | | 1 | |
| Poor (≥10% tumour) | **1.551** | **1.147 to 2.099** | **1.748** | **1.236 to 2.471** | **1.472** | **1.29 to 1.679** | **2.453** | **1.561 to 3.855** | 0.872 | 0.645 to 1.179 |
| Time effect | | | | | **0.979** | **0.967 to 0.992** | | | | |
| Excision | | | | | | | | | | |
| Wide/radical | 1 | | 1 | | 1 | | | | 1 | |
| Marginal | | | **0.379** | **0.221 to 0.651** | 1.089 | 0.929 to 1.277 | | | 0.811 | 0.442 to 1.490 |
| Intralesional | | | **0.331** | **0.163 to 0.673** | 1.401 | 0.975 to 2.014 | | | **5.296** | **2.856 to 9.819** |
| Location | | | | | | | | | | |
| Other | 1 | | 1 | | 1 | | | | | |
| Proximal femur/humerus | **2.299** | **1.585 to 3.335** | **2.717** | **1.696 to 4.354** | **1.326** | **1.151 to 1.527** | | | | |
| Axial | **6.919** | **4.497 to 10.65** | **11.535** | **6.113 to 21.767** | **1.335** | **1.016 to 1.755** | | | | |
| Sex | | | | | | | | | | |
| Female | 1 | | 1 | | 1 | | 1 | | 1 | |
| Male | 0.986 | 0.733 to 1.326 | 1.305 | 0.923 to 1.844 | **1.589** | **1.418 to 1.782** | **1.823** | **1.302 to 2.552** | 1.165 | 0.836 to 1.623 |
| Volume | | | | | | | | | | |
| <200 | 1 | | 1 | | 1 | | 1 | | 1 | |
| ≥200 | 1.093 | 0.772 to 1.548 | 0.756 | 0.516 to 1.106 | 1.03 | 0.915 to 1.16 | 1.13 | 0.804 to 1.589 | 0.866 | 0.591 to 1.271 |

Bold values indicate hazard ratios (HRs) with 95% confidence intervals (CIs) that exclude the null.

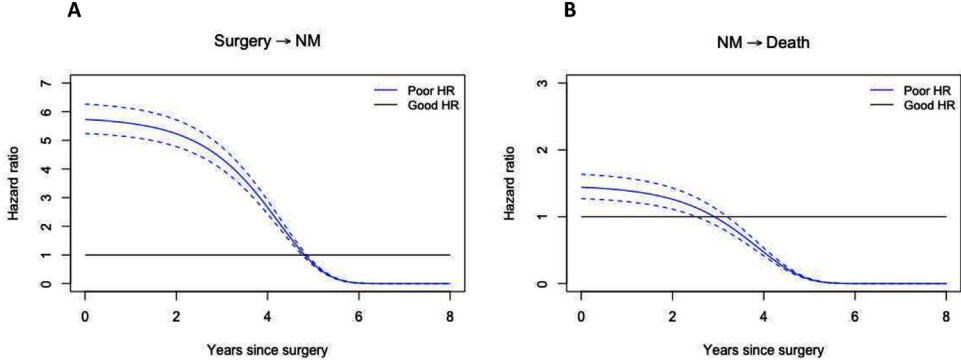

**Figure 3** Time-varying hazard for histological response. (A) Hazard for transitioning from surgery to new metastatic disease (NM). (B) Hazard for transitioning from NM to death. *Blue*: poor histological response; *black*: good histological response; *dashed line*: pointwise CI for the HR of poor histological response.

>18 years experienced a decreased risk of SM (HR=0.58, 95% CI 0.36 to 0.91), NM2 (HR=0.67, 95% CI 0.52 to 0.87), death after LR (HR=0.35, 95% CI 0.21 to 0.56), death after LR+NM (HR=0.47, 95% CI 0.31 to 0.71), and death after NM2 (HR=0.53, 95% CI 0.34 to 0.83).

Male sex was associated with an increased risk of LR+NM (HR=1.40, 95% CI 1.08 to 1.82), NM2 (HR=1.34, 95% CI 1.09 to 1.65), death after NM (HR=1.59, 95% CI 1.42 to 1.79), and death after LR+NM (HR=1.82, 95% CI 1.30 to 2.55), but a decreased risk of SM (HR=0.72, 95% CI 0.53 to 0.98).

A tumour volume of >200 was associated with a modest increase in risk of NM (HR=1.51, 95% CI 1.40 to 1.64) and LR+NM (HR=1.50, 95% CI 1.15 to 1.95), and a decrease in risk of NM2 (HR=0.76, 95% CI 0.59 to 0.98).

### Time-varying effect

Histological response violated the proportional hazards assumption for the transition surgery to NM and NM to death. Therefore, the effect of histological response is modelled as function of time. The HR of transitioning to NM decreased with time, initially slowly, and more quickly around the 3-year mark (figure 3A). For the transition from NM to death, the decrease was less pronounced (figure 3B). The time-varying HR can be computed for any given timepoint using the formula shown below. For example, the HR for the transition from surgery to NM is given by:

$$\text{HR} = \text{constant} \times \text{time-varying-effect}^{\exp(t)}.$$

Here, *constant* is the surgery to NM HR of 5.81, the *time-varying effect* is 0.986 (table 2), and *t* is time in years. At 1 year, the HR is: $\text{HR} = 5.81 \times 0.985^{\exp(1)} = 5.76$. At 3 years, the HR has decreased to $5.81 \times 0.985^{\exp(3)} = 4.36$.

### State-occupation probability plots

Figure 4 illustrates, for a set of example patients, osteosarcoma disease progression through state occupation probability plots, where the probability of being in a given state is plotted against time, in years since surgery. Whereas HRs only inform on the relative change in risk, state occupation probabilities show the absolute probability of being in a given state (ie, having experienced an intermediate event) over a period of time. Panels 1A–1C

illustrate the effect of age category, with in panel 1A the state occupation probabilities of a patient aged 12–18, with all other characteristics reference categories (good histological response, wide/radical excision, tumour volume <200 cm³, female, tumour location of category 'other'), and in panels 1B and 1C the state occupation probabilities of patients aged <12 and ≥18, respectively. Patients aged 12–18 were least likely to remain event free in the state of surgery (green) and had a moderately higher probability of death (purple). Patients aged <12 were most likely to remain event free with a lower probability of an intermediate event or death. Patients aged >18 were at greater risk of NM (blue) than both other age groups.

When compared with the reference patient (1A), patients with an axial osteosarcoma (2A) were at lower risk of NM, higher risk of death, and a much-increased risk of LR (yellow), and were most likely to experience the latter event 1–3 years after surgery. An intralesional surgical excision (2B), in contrast, resulted an in increased risk of LR that remained relatively constant over time. Patients with a poor histological response (2C) were more likely to experience NM, or an SM (red). Panels 3A–3C and 4A–4C show the state occupation probabilities for patients who experienced an NM or LR, respectively, at 1, 2 and 4 years after surgery. The later the event occurred, the lower the probability of the disease progressing to death. For example, for LRs 1 year and 4 years after surgery, the 4-year survival probabilities were approximately 0.65 (4A) and 0.8 (4C), respectively.

### DISCUSSION

This study is unique in estimating the hazard of transitions to and from different intermediate events, allowing for a better insight into the prognosis of specific patient groups than conventional analyses that consider only the overall effect of prognostic factors. Using a metastasis-free subset of the EURAMOS-1 data,[6] we interpreted the effect of six prognostic factors on osteosarcoma disease progression after surgery in a multi-staged way. Multistate model estimation is in practice limited to transitions with a

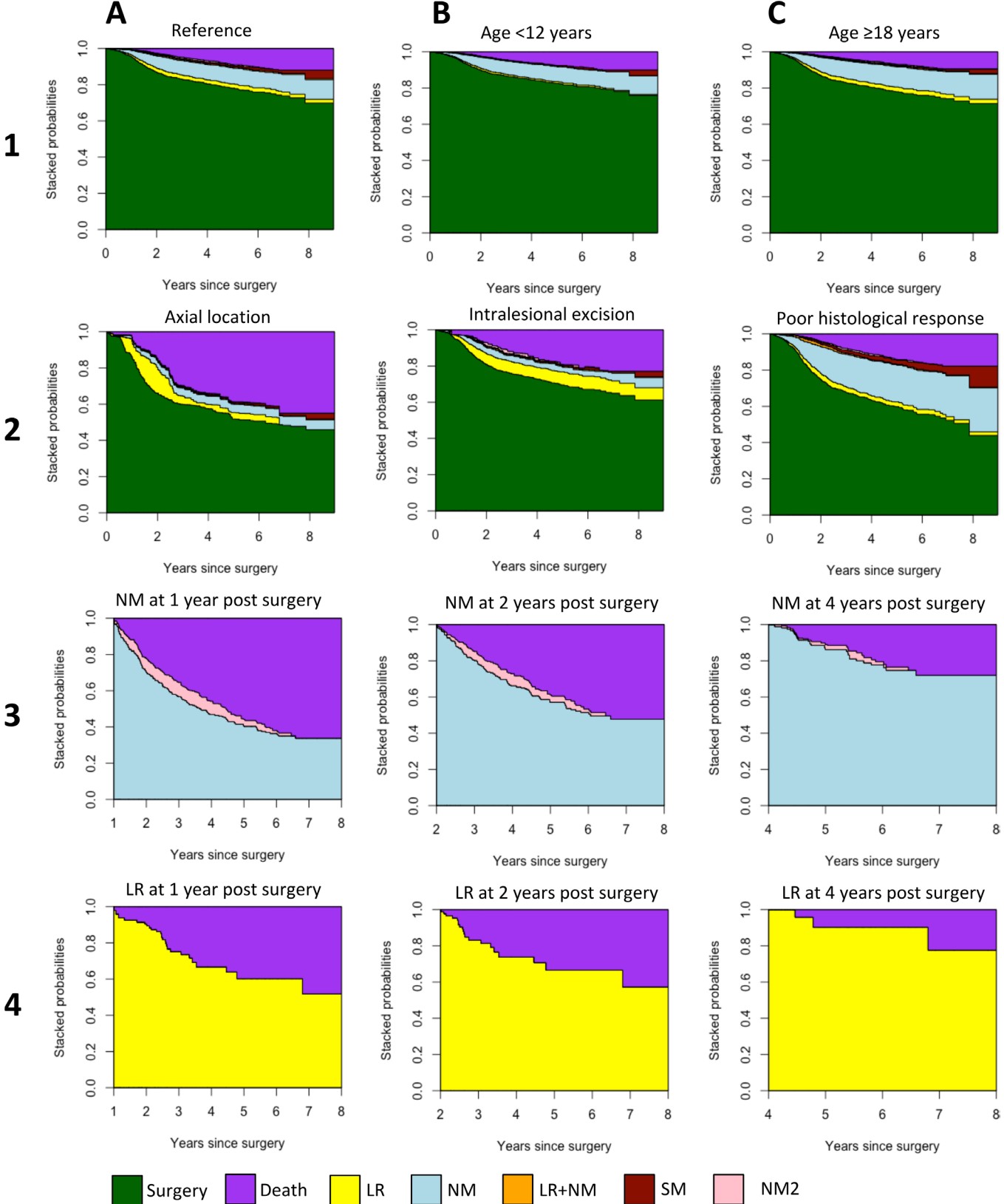

**Figure 4** Stacked state occupation probabilities for patients with different characteristics. Patient characteristics are defined with respect to the reference patient, shown in *1A*: patient with reference characteristics: age 12–18 years, good histological response, wide/radical excision, tumour volume <200 cm³, female, tumour location of category 'other'. *1B*: patient aged <12 years. *1C*: patient aged >18 years. *2A:* patient with axial tumour location. *2B*: patient with poor histological response. *2C*: patient with intralesional excision. *3A–3C*: patient with reference characteristics experiencing a new metastatic disease (NM) at 1 year (*3A*), at 2 years (*3B*), and at 4 years (*3C*). *4A–4C*: patient with reference characteristics experiencing local recurrence (LR) at 1 year (*3A*), at 2 years (*3B*), and at 4 years (*3C*). SM, secondary malignancy.

sufficient number of events per predictor category. Categories with few patients (eg, axial tumour location and intralesional surgical excision), when combined with rare intermediate events (eg, NM+LR and SM), will preclude statistical inference. Specifically, we were unable to estimate the effect of prognostic factors surgical excision and tumour location for four and five out of a total of 10 transitions, respectively.

The most notable adverse risk factor was an axial location of the osteosarcoma. A previous analysis of the EURAMOS-1 data found moderate HRs of 1.74 (95% CI 1.06 to 2.85) and 1.29 (95% CI 0.86 to 1.95) for OS and EFS, respectively.[6] In our results, an axial location was chiefly relevant to the transition from surgery to LR and LR to death, yielding substantially higher transition-specific HRs of 10.84 (95% CI 8.45 to 13.86) and 11.54 (95% CI 6.11 to 21.77), respectively. In contrast, the occurrence of an NM was less common (HR=0.70, 95% CI 0.56 to 0.87). Clinically, this may have implications for additional treatment and follow-up for osteosarcomas at this site.

A poor histological response has widely been shown to adversely affect OS.[6 12–14] The previous EURAMOS-1 analysis found HRs of 2.14 (95% CI 1.76 to 2.58) and 2.45 (95% CI 1.88 to 3.20) for EFS and OS, respectively.[6] In our study, a poor histological response increased the risk of NM nearly 6× (HR 5.81, 95% CI 5.31 to 6.36) and the risk of LR+NM nearly 4× (HR 3.81, 95% CI 2.86 to 5.10), was generally predictive of death after an immediate event, but only modestly affected the risk of an event-free death. In particular, the risk of NM was substantially increased during the first 2.5 years, but the predictive strength steeply decreased after 3 years since surgery, with the HR nearly halved at 4 years of follow-up.

For an intralesional excision, compared to a wide/radical excision, HRs of 2.73 (95% CI 1.15 to 6.47) and 1.98 (95% CI 0.91 to 4.30) were previously found for OS and EFS, respectively.[6] Our results confirm a higher probability of an LR after intralesional surgery (HR 2.08, 95% CI 1.32 to 3.29), with this risk remaining comparatively constant during follow-up. Our study shows that developing recurrent NM is a risk factor of intralesional surgery (HR 4.10, 95% CI 2.61 to 6.43), with the risk of a subsequent poor outcome increased more than 5× (HR 5.30, 95% CI 2.86 to 9.82), which is substantially higher than the previously reported HR for OS.

The remaining variables in our model had more modest HRs, which varied less across transitions. Previously, male sex was found to have only a modest adverse effect on EFS and OS.[6 15] We observed an increased risk of LR+NM and NM2, and subsequent death, but a decreased risk of SM. For greater tumour volume, an EFS HR of 1.24 (95% CI 1.00 to 1.52) was previously found.[6] We observed an increased risk for the events NM and LR+NM, with no significant effect for any transition to death, which is in line with previous results for OS (HR=1.19, 95% CI 0.92 to 1.55).[6]

Young age was found to be protective for OS and EFS in the previous EURAMOS-1 analysis[6] and other studies.[15 16]

However, in Collins' study,[15] the hazard lost its significance for survival at 2 years post surgery, and our study did not confirm increased risk of event. We found that both the youngest patients (<12 years) and those aged >18 years were protected from experiencing an event. Given the occurrence of LR, LR+NM or NM2, the probability of poor outcome was highest in the young age group, whereas patients aged >18 years had a remarkably better outcome. This may have implications for the treatment of LR in younger versus older patients, suggesting that older patients could be treated by local treatment only, whereas for younger patients, a more intensified treatment regimen with additional systemic treatment should be considered.

## CONCLUSION

Using data of more than 1600 patients from the EURAMOS-1 trial, we estimated a multistate model with intermediate events. This study shows the added value of considering prognostic factors specific to transition and in light of event history. Our findings indicate that young patients with an LR have a poor prognosis, suggesting that it may be beneficial to investigate additional treatment options for this subgroup. Additionally, our results stress the necessity of increased monitoring of patients with axial tumours for LRs, and patients with a poor histological response for NM, while noting for the latter that predictive power decreases over time. We show that a multistate model yields additional clinical knowledge for specific osteosarcoma patient groups, when compared to conventional OS and EFS analyses. Previously, multistate models have been used to model disease progression in soft-tissue sarcoma,[17 18] Ewing sarcoma[19] and breast cancer.[20] Reanalysing data from other large randomised studies, using the multistate approach, may also yield valuable insights for patients with other oncological conditions.

**Author affiliations**
[1]MRC Integrative Epidemiology Unit, Bristol Medical School, University of Bristol, Bristol, UK
[2]Population Health Sciences, Bristol Medical School, University of Bristol, Bristol, UK
[3]Department of Biomedical Data Science, Leiden University Medical Center, Leiden, Netherlands
[4]Department of Medical Oncology, Leiden University Medical Center, Leiden, Netherlands
[5]Mathematical Institute, Leiden University, Leiden, Netherlands
[6]Department of Solid Tumours, Princess Máxima Centre, Utrecht, Netherlands
[7]Department of Orthopaedics, Leiden University Medical Center, Leiden, Netherlands
[8]Cancer Institute, Faculty of Medical Sciences, University College London, London, UK

**Contributors** A-DH: writing–original draft preparation, visualization, formal analysis, software. CL: data curation, writing–review and editing. JA: writing–review & editing. MvdS: writing–review and editing. JW: investigation. HG: conceptualisation, writing–review and editing. MF: guarantor, conceptualisation, supervision, writing–review and editing.

**Funding** We thank the EURAMOS-1 trial investigators and participants.This work was supported by the Dutch Foundation KiKa (Stichting Kinderen Kankervrij), grant 163, through the project Meta-analysis of individual patient data to investigate dose-intensity relation with survival outcome for patients with osteosarcoma. A-DH

was supported by Integrative Epidemiology Unit, which receives funding from the UK Medical Research Council and the University of Bristol (MC_UU_00011/3).

**Competing interests**  None declared.

**Patient consent for publication**  Not applicable.

**Ethics approval**  All participating institutions obtained regulatory and ethics approvals in accordance to their national rules and regulations. Written informed content was obtained from all trial participants or legal guardians before beginning protocol therapy. The protocol was approved by the Local Research Ethics Committee (LREC) in Gent (coordinating Ethics Committee for Euramos, Belgium), the Central Committee on Research Involving Human Subjects (CCMO) in the Netherlands, the LREC in Leiden (the Netherlands), and the Multi-Centre Research Ethics Committee (MREC) and LREC in the UK. Note that our study is a reanalysis of an already published trial. Participants gave informed consent to participate in the study before taking part.

**Provenance and peer review**  Not commissioned; externally peer reviewed.

**Data availability statement**  Data may be obtained from a third party and are not publicly available. A request to access the EURAMOS-1 trial data may be submitted to the MRC Clinical Trials Unit (CTU, London). The application requires completion of an analysis and data release request form, where the applicant provides a project summary (detailing the motivation of the data request, the background and objectives of their project, and the reasons for requesting this specific dataset), the data requirements (for this study, the anonymised individual-level data for all registered patients were requested, including (demographic) patient characteristics, disease characteristics, pathology and surgical information, treatment data, and major events), and details on the proposed publication, authorship and acknowledgements policy. Data applications are submitted to the Coordinating Data Center (CDC, London), and subject to review by the Trial Management Group and the Trial Steering Committee.

**ORCID iDs**
Audinga-Dea Hazewinkel http://orcid.org/0000-0002-6923-4388
Jeremy Whelan http://orcid.org/0000-0001-6793-5722

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
