## [Reviewer comments · BMJ Open]

ARTICLE DETAILS

TITLE (PROVISIONAL)	Disease progression in osteosarcoma: a multi-state model for the EURAMOS-1 (European and American Osteosarcoma Study) randomised clinical trial
AUTHORS	Hazewinkel, Audinga-Dea; Lancia, Carlo; Anninga, Jakob; van de Sande, Michiel; Whelan, Jeremy; Gelderblom, Hans; Fiocco, Marta

VERSION 1 – REVIEW

REVIEWER	Gaspar, Nathalie Gustave Roussy, Department of Oncologie for child and adolescent
REVIEW RETURNED	20-Oct-2021

GENERAL COMMENTS	The authors wanted to investigate the effect of prognostic factors in a multi-state framework on survival in a large population of osteosarcoma patients. Of interest is how prognostic factors affect different disease progression stages after surgery, with stages of local recurrence (LR), new metastatic disease (NM), local recurrence + new metastatic malignancy, a second new metastatic disease and death. Statistic should be reviewed by a statistician Population selection - Why to limit to patient with localized disease and operated, and exclude metastatic patients, not operable patients and progression before surgery?- In the exclusion category Figure 1 : please explain what areo 22 patients with disease-dependant non-randomisation?o 170 patients with progress of metastatic disease as first event but no record of pulmonary/other metastasis at admission ?- Author excluded “ patients from whom progression of new metastatic disease was found, but no new metastatic disease was recorded” what does ot means? Is it the same category than above ? Patient demographic - Why the cut-off age was set up at 12 and 18 years old?- Why to choose initial tumour volume as variable while tumour size if the factor usually described in osteosarcoma studies?- Why separate proximal femur/humerus from other long bone location?- Can you specify which axial tumour were included as they have been operated? Rib ? mandibules??- details on the treatment really received is missing e.g. type of surgery ? Conclusion The authors suggest that young patients with a local recurrence may benefit from a more intensive treatment regimen, and stress
--

	the necessity of increased monitoring of patients with axial tumours for local recurrences, and patients with a poor histological response for new metastatic disease, while noting for the latter that predictive power decreases over time. - “young patients with a local recurrence may benefit from a more intensive treatment regimen” nothing is said about the treatment they received either type of chemo or surgery, is it a more aggressive regimen or more aggressive surgery that is needed?
--	--

VERSION 1 – AUTHOR RESPONSE

Reviewer comments

1. Why to limit to patient with localized disease and operated, and exclude metastatic patients, not operable patients and progression before surgery?

Our research question is regarding the effect of various predictors on prognosis after surgery, which excludes patients who did not have surgery. Patients with non-operable disease, metastatic and progression before surgery are a biologically distinct group with a much poorer prognosis. Patients whose initial presentation included clinically detectable metastatic disease continue to have clinically detectable metastatic disease after definitive surgery. Some of those patients go on to a complete remission or state of minimal disease following resection of clinically detectable metastatic disease. Other patients will never achieve complete remission because surgical resection of metastatic disease fails. These patients are extremely distinct from the patients who did not present with clinically detectable metastatic disease and who are in a state of complete remission/ minimal residual disease following surgery. Our patient exclusion criteria were in the interest of obtaining a homogeneous population. We appreciate this was not clearly communicated in the paper, and have now clarified this in the text (please see paragraph 1, page 5) :

“To ensure valid inference, we selected a homogeneous subset of the data, excluding patients with a non-resectable primary tumour and patients with clinically detectable metastatic disease prior to surgery, as the latter comprise a biologically distinct population with a much poorer prognosis.”

2. In the exclusion category Figure 1 : please explain what are
 - a) 22 patients with disease-dependent non-randomisation?
 - b) 170 patients with progress of metastatic disease as first event but no record of pulmonary/other metastasis at admission?

a) A small number of patients, who were initially included in the study, were later found to be ineligible at randomization. Disease-dependent non-randomisation pertains to 22 patients, 11 of which were excluded because of progression of metastatic disease or new metastatic disease, while for the remaining 11 primary and/or metastatic disease was unresectable. We have now. Clarified this in the legend of Figure 1 (please see paragraph 1, page 17):

“; b) 22 randomised patients were later found to be ineligible due to progression of metastatic disease or new metastatic disease (n=11), or primary and/or metastatic unresectable disease (n=11)”

b) 357 patients had metastases at registration. A further 170 patients had no record of metastases at registration but were later found to have progression of metastatic disease after surgery (without

new metastatic disease being recorded prior to this). For progression to occur, metastatic disease must be already present, so these patients were retrospectively reclassified as having metastases prior to surgery. Smeland et al. take the same approach in their 2019 EURAMOS article (page 29, second column below Fig. 1): [https://www.ejancer.com/article/S0959-8049\(18\)31534-X/fulltext](https://www.ejancer.com/article/S0959-8049(18)31534-X/fulltext)
We have clarified this in the legend of Figure 1 (please see paragraph 1, page 17):

“a) to ensure a homogeneous study population, we excluded patients with metastases prior to surgery. For 357 patients metastases were recorded at registration, while for 170 patients, progression of new metastatic disease was found after surgery, while no metastases were recorded at registration. These patients were retrospectively reclassified as having metastatic disease prior to surgery and excluded from the analysis”

We have also amended Figure 1, replacing ‘admission’ with ‘registration’, as we noticed that we were using those terms interchangeably, which may be confusing.

3. Author excluded “patients from whom progression of new metastatic disease was found, but no new metastatic disease was recorded” what does this mean? Is it the same category as above?

Yes, it's the same. We have rephrased our description of this exclusion category in the text, please see our response to comment 2.b) above.

4. Why was the cut-off age set up at 12 and 18 years old?

We wished to be consistent with previous publications on the EURAMOS study (namely, the 2019 Smeland paper, that looks at overall and event-free survival), in order to facilitate an informative results comparison. Smeland et al. (2019) used these same cut-off points ([https://www.ejancer.com/article/S0959-8049\(18\)31534-X/fulltext](https://www.ejancer.com/article/S0959-8049(18)31534-X/fulltext))

5. Why choose initial tumour volume as a variable while tumour size is the factor usually described in osteosarcoma studies?

Both volume and size are used in sarcoma studies. For example, Ziegele et al. (2016) found that volume was a better predictor than tumour size in soft tissue sarcoma, while Kim et al. (2015), for osteosarcoma, found no preferred option when comparing the associations of absolute tumour length, absolute tumour volume, and relative tumour volume with survival. As such, we felt there was no clear consensus on the best choice of predictor in this category, and went forward with volume as a measure. While tumour size (largest dimension in one plane) is indeed a more frequent choice for osteosarcoma, our results w.r.t. volume were in accordance with previous results found, in context of both overall survival and event-free survival (Smeland et al., 2019).

Ziegele et. al. (2016): <https://pubmed.ncbi.nlm.nih.gov/26909140/>

Smeland et al. (2019): [https://www.ejancer.com/article/S0959-8049\(18\)31534-X/fulltext](https://www.ejancer.com/article/S0959-8049(18)31534-X/fulltext)

Kim et al. (2015): <https://bmccancer.biomedcentral.com/articles/10.1186/s12885-015-1129-9>

6. Why separate proximal femur/humerus from other long bone location?

A previous study found that a proximal location of the tumour in the femur/humerus appeared to have poorer outcomes than those arising in distal long bone locations (Cates & Schoenecker,

2016; <https://pubmed.ncbi.nlm.nih.gov/27145235/>). Additionally, as for the age cut-off (see comment 4), we wished to be consistent with the categorization used in Smeland's 2019 paper.

7. Can you specify which axial tumour were included as they have been operated? Rib ? mandibules??

The protocol distinguished between axial tumours in the pelvis/sacrum, rib, and spine (only 1 incidence, and not present in our subset of patients). In our study, the axial tumours comprise those of the pelvis/sacrum (41) and rib (14). We have clarified this in the footnotes of Table 1 (please see page 6):

“d) Tumour location was defined in accordance with the definition used in the Smeland et al. (2019) analysis of survival and prognosis in the EURAMOS-1 trial. Information was pooled from study variables 'site' (eg. femur, pelvis, spine, etc.) and 'location' (e.g. proximal, axial, etc.). Observed axial tumour locations included rib (14) and pelvis/sacrum (41).”

8. Details on the treatment really received is missing e.g. type of surgery?

We have now added detail on the treatment the patients were randomised to (please see last paragraph page 4):

“Resection of the primary tumour was performed post neoadjuvant chemotherapy. Following surgery, 1136 of 2260 patients were randomised to treatment, subject to the histological response as assessed in the resected specimen. Patients with a poor response ($\geq 10\%$ viable tumour) were allocated MAP or MAP with ifosfamide and etoposide, while patients with a good response ($< 10\%$) received MAP or MAP followed by pegylated interferon. The primaranalysis found no beneficial effect of experimental treatment in either group.”

The type of surgery performed is reflected in one of our prognostic variables, surgical excision (see Table 1). There, we distinguish between wide or radical excisions, marginal excisions and intralesional excisions

9. The authors suggest that young patients with a local recurrence may benefit from a more intensive treatment regimen, and stress the necessity of increased monitoring of patients with axial tumours for local recurrences, and patients with a poor histological response for new metastatic disease, while noting for the latter that predictive power decreases over time. - “young patients with a local recurrence may benefit from a more intensive treatment regimen” nothing is said about the treatment they received either type of chemo or surgery, is it a more aggressive regimen or more aggressive surgery that is needed?

What we meant to say here is that it may be a good idea to consider potential alterations to treatment for young patients, as they fare so poorly if they experience a local recurrence after surgery. We agree that the statement in the manuscript is worded too strongly and not appropriate as is. We have rephrased it (please see final paragraph, page 14):

“Our findings indicate that young patients with a local recurrence have a poor prognosis, suggesting that it may be beneficial to investigate additional treatment options for this subgroup. Additionally, our results stress...”

VERSION 2 – REVIEW

REVIEWER	Gaspar, Nathalie Gustave Roussy, Department of Oncologie for child and adolescent
REVIEW RETURNED	06-Dec-2021

GENERAL COMMENTS	The authors investigate the effect of prognostic factors in a multi-state framework on survival in a large population of osteosarcoma patients issued from EURAMOS-1 trial, highlighting how prognostic factors could affect different disease progression stages after surgery, with stages of local recurrence (LR), new metastatic disease (NM), local recurrence + new metastatic malignancy, a second new metastatic disease and death. The changes made to the manuscript answer my previous questions. If the statistical review is Ok for statistician, the manuscript should be accepted.
--

REVIEWER	Xu, Jianming Baylor College of Medicine
REVIEW RETURNED	24-Jan-2022

GENERAL COMMENTS	In this manuscript, the authors re-analyzed the clinical patient data of the previously reported EURAMOS-1 study with a large cohort using a multi-state model. In this model, the authors used the surgery point as starting state and defined a total of 7 possible states and 10 transitions according to the available patient data follow-up and osteosarcoma progression. From the massive data analyses, the authors concluded that patients with axial tumors should be monitored for local recurrence; patients with the poor histological response should be monitored for the development of new metastatic disease; young patients (<12) with a local recurrence should receive additional treatment options. Overall, the source of the patient data, the statistical analysis procedures, and the results are clearly described. The outcomes should have important implications in improving the accuracy and sensitivity of risk estimation for osteosarcoma patients. Minor comment: In line 1 on page 7, please define what is the “MAP” treatment.
--